# A Study of the Temperature-Dependent Surface and Upper Critical Magnetic Fields in KFeSe and LaSrCuO Superconductors

Suppanyou Meakniti [1,*], Pongkaew Udomsamuthirun [1], Arpapong Changjan [2], Grittichon Chanilkul [1] and Thitipong Kruaehong [3]

1  Department of Physics, Faculty of Science, Srinakharinwirot University, Bangkok 10110, Thailand
2  Department of Environmental Technology for Agriculture, Faculty of Science and Technology, Pathumwan Institute of Technology, Bangkok 10330, Thailand
3  Department of Industrial Electrics, Faculty of Science and Technology, Suratthani Rajabhat University, Surat Thani 84100, Thailand
*  Correspondence: suppanyou.fern@gmail.com

**Abstract:** The critical magnetic field is one of the most interesting properties of superconductors. Thus, this study aimed to investigate the surface and upper critical magnetic fields of superconductors in Fe-based and cuprate superconductors as KFeSe and LaSrCuO superconductors, respectively. The anisotropic two-band Ginzburg–Landau method was used to generate the analytic equation. The analytics were shown for the simplified equation so that a second-order polynomial temperature-dependent equation could be applied and fitted to the experimental results of KFeSe and LaSrCuO superconductors. After that, numerical calculations were applied to find the shape of the Fermi surface, which is an important component within the band structure. It was found that the anisotropy of the Fermi surface for each band structure was affected by the upper critical magnetic field and the surface critical magnetic field to the upper critical magnetic field of the superconductors. The second-order polynomial temperature-dependent model can be applied to other superconductors to predict the surface and upper critical magnetic fields.

**Keywords:** surface critical magnetic field; upper critical magnetic field; Ginzburg–Landau theory





## 1. Introduction

A superconductor is a material that has no resistance at a low temperature, known as the critical temperature. It is influenced by three factors, including the critical temperature, critical current density, and critical magnetic field that can convert a superconducting state to a normal state. There are three kinds of critical magnetic fields: lower, upper, and surface. The maximum surface magnetic field to convert a superconductor into a normal conductor has a greater value than the upper critical magnetic field, which is known as the surface critical magnetic field [1–3]. It was intended that the investigation of the surface critical magnetic field properties would offer recommendations for producing thin film superconductors. It was intended that the investigation of the surface critical magnetic field properties would offer recommendations for producing thin film superconductors and make superconductors more efficient. Superconductors fall into two major categories, namely type I and type II superconductor [1]. For type I superconductors, the superconductors lose superconductivity and convert to normal conductors when the external magnetic fields reach the critical level. Type II superconductors have two critical magnetic fields, called the lower critical magnetic field ($H_{c1}$) and the upper critical magnetic field ($H_{c2}$). However, the critical magnetic field of isotropic one-band superconductors near the surface of materials is greater than the upper critical magnetic field of bulk superconductors. The superconductors lose superconductivity but remain at the surface when the external magnetic fields reach the upper critical magnetic field. They will then lose superconductivity

and convert to normal conductors when the external magnetic fields reach the surface critical magnetic field ($H_{c3}$) [2,3]. In alloy superconductors, the calculation of the Ginzburg–Landau approach was done and applied to the alloy superconductor. A minor effect of the thickness of the film on the surface critical field was found to be $H_{c3} \approx 1.7 H_{c2}$ [4]. However, the derivation of this relation can be found to be dependent on many parameters such as temperature dependence and Ginzburg–Landau parameters ($\kappa$) [5,6]. Cuprate superconductors with critical temperatures as high as 35 K have been discovered [7]. When pressure is added to the system, the critical temperature of the cuprate superconductor rises to 50 K, as measured by the research group [8]; when Y atoms replace La atoms (LaBaCuO to YBaCuO), the temperature climbs to 92 K. Regardless of a high critical temperature, however, this category of superconductors has a low critical magnetic field. In LaSrCuO superconductors, the anisotropy of critical field was found; $\frac{H_{c3}}{H_{c2}} = 1.8$ for c-crystal and $\frac{H_{c3}}{H_{c2}} = 4.0$ for a-crystal [9]. LaSrCuO, also known as LSCO, is a complex oxide compound made up of the elements lanthanum (La), strontium (Sr), copper (Cu), and oxygen (O). It belongs to the family of high-temperature superconducting materials known as cuprates. To ensure a surplus amount of 3 mol% CuO and $x$ mol% Sr ions, the proportion of CuO and Sr ions to the required amount in a full reaction of La$_2$Sr$_x$CuO$_4$ was adjusted. This resulted in the reaction scheme $1.03 CuO + (2 - x)/2\, La_{2-x}O_3 + x\, Sr_xCO_3 \rightarrow La_{2-x}Sr_xCuO_4$ [10]. LSCO can be grown as a single crystal using techniques such as the floating zone method or the flux method [11]. Single crystals of LSCO typically have a layered structure, with layers of CuO$_2$ planes separated by layers of LaO or SrO. The CuO$_2$ planes are critical to the superconducting properties of the material. LSCO exhibits superconductivity at temperatures below about 40 K, which LaSrCuO has a critical temperature of about 35 K [9]. The properties of LSCO can be tuned by varying the composition or doping the material with other elements. Through extensive research, it has been determined that the ideal level of doping with Sr$^{2+}$ ions is $x = 0.16$, which produces the highest $T_c$. The precise amount of doping and the range of $x = 0.10$ to $x = 0.18$ have been thoroughly investigated in reference [10]. The crystal structure of LaSrCuO belongs to the tetragonal system with space group I4/mmm [12]. The thermal Hall effect found in La$_{2-x}$Sr$_x$CuO$_4$ [13–15] has been linked to chiral phonon excitations that require specific crystal structures. Although the orthorhombic space group *Bmab* (space group 64) provides a good average description of the La$_{2-x}$Sr$_x$CuO$_4$ structure [16,17], there is growing evidence of subtle structural distortions in both doped and undoped samples. The available literature suggests that charge stripe order in LSCO is indicated in tetragonal notation, while deviations from space group 64 are best described in orthorhombic notation [18]. Iron-based superconductors have been discovered [19]. It is highly fascinating due to its exceptionally high upper critical magnetic field and its use in a solid-state reaction to create LaFeAsO superconductors. The upper critical magnetic field was found to be as high as 55 T [20]. The $\frac{H_{c3}}{H_{c2}}$ of lead was measured and found to have different values in the vicinity of the critical temperature [21]. Recently, for Fe-based superconductors, an experimental investigation on the magnetic characteristics of the K$_{0.73}$Fe$_{1.68}$Se$_2$ superconductor was carried out, in which the result of the ratio $H_{c3}$ to $H_{c2}$ was about 4.4 [22]. The Fe-based superconductors AFeCh, where A is an alkali metal (e.g., K) and Ch is a chalcogen (e.g., Se), were discovered by [23,24] and are referred to as iron chalcogenide superconductors (FeCh). They have the same structure as the iron-pnictide 122 system [25–27]. The single crystalline KFeSe materials consist of layers of FeSe and K atoms stacked on top of each other in a body-centered tetragonal structure with space group I4/mmm. At Neel temperatures around $T_N \approx 140$–150 K and 520–550 K, the Fe-Se layers in the material display antiferromagnetic order in three dimensions, with Fe$^{2+}$ moments oriented along the c axis. Fe-Se based materials undergo a structural transition during their magnetic transition, where for Fe-Ch materials, the transition is from a tetragonal (I4/mmm) to another tetragonal structure (I4/m). These materials can exhibit superconductivity at a critical temperature of around 30–33 K while retaining antiferromagnetism [22]. The crystal structure of KFe$_2$Se$_2$ can be represented as a stacking of K sheets and [Fe$_2$Se$_2$] blocks. Specifically, the K atoms are located at 2a (0, 0, 0),

Fe atoms at 4d (0, 1/2, 1/4), and Ch (Se) atoms at 4e (0, 0, $z_{Ch}$), where $z_{Ch}$ represents the internal coordinate. The structural parameters of $KFe_2Se_2$ include lattice constants (a, c in angstrom unit), internal coordinate ($z_{Ch}$), bond length (d in angstrom unit), bond angles ($\Theta$ in degrees unit), and anion height ($\Delta z$ in angstrom unit), which have been presented by. The $KFe_2Se_2$ is characterized by its energy bands, Fermi surfaces, and densities of states (DOS). The near-Fermi bands display a complex "mixed" nature, consisting of both quasi-flat bands along the Γ-Z axis and a set of highly dispersive bands that intersect the Fermi level [28]. Overall, the crystal structure of KFeSe is characterized by its layered structure, with the FeSe layers separated by the K atoms. This layered structure is important for its electronic properties, such as high-temperature superconductivity. From the theoretical view, the two-band Ginzburg–Landau (GL) theory was used to determine the surface critical magnetic field of the $MgB_2$ superconductor [29]. The two-band GL free energy was taken by the minimization process and a ratio of $H_{c3}/H_{c2}$ was found with the same as a well-known formula, $H_{c3}(T) = 1.66H_{c2}(T)$. In 2014, Meakniti et al. [30] used a one-band Ginzburg–Landau method to study the effects of diamagnetism, ferromagnetism, antiferromagnetism, and paramagnetism on the surface critical magnetic field of a layered magnetic superconductor. They discovered that paramagnetism and antiferromagnetism from $H_{c3}$ to $H_{c2}$ were nearly equivalent to 1.66. In 2017, Changjan and Udomsamuthirun [31] studied the four temperature-dependent models of the surface critical magnetic field of isotropic single-band superconductors applied to lead-bismuth superconductors.

In this research, the surface critical magnetic fields and upper critical magnetic fields of two-band magnetic superconductors were conducted, which included temperature and anisotropy-dependent, concentrating on the second-order polynomial temperature functions as well as the anisotropic functions by Hass and Maki, and Posazhennikova. The numerical calculation by the iteration method was used for fitting the theoretical parameters and experimental parameters. The application of temperature and anisotropy dependence was made to describe the experimental data of iron-based and cuprate superconductors as KFeSe and LaSrCuO superconductors, respectively.

## 2. Model and Calculations

This study concentrated on the anisotropic dependence and temperature dependence of two-band magnetic superconductors [32–34]. The two-band Ginzburg–Landau free-energy function with two order parameters on $\psi_1$ and $\psi_2$ can be written as

$$F_{sc}[\psi_1, \psi_2] = \int d^3r \left( F_1 + F_2 + F_{12} + \gamma_0 + \gamma_1 B + \gamma_2 \frac{B^2}{2\mu_0} \right) \tag{1}$$

$$F_{i(i=1,2)} = \frac{1}{2m_i} \left| \left( -i\hbar\nabla - 2e\vec{A} \right)\psi_i \right|^2 \left\langle f_i^2(\hat{k}) \right\rangle + \alpha_i(T)\psi_i^2 \left\langle f_i^2(\hat{k}) \right\rangle + \frac{1}{2}\beta_i\psi_i^4 \left\langle f_i^2(\hat{k}) \right\rangle \tag{2}$$

$$F_{12} = \varepsilon\left(\psi_1^*\psi_2 + c.c.\right)\left\langle f_1(\hat{k})f_2(\hat{k}) \right\rangle$$

$$+\varepsilon_1 \left\{ \left( i\hbar\nabla - 2e\vec{A} \right)\psi_1^* \left( -i\hbar\nabla - 2e\vec{A} \right)\psi_2 + c.c. \right\}\left\langle f_1(\hat{k})f_2(\hat{k}) \right\rangle \tag{3}$$

where $F_i$ represents the individual bands' anisotropic free energies, $F_{12}$ represents the interaction between the first and second band, $m_i$ stands for the carriers' effective mass, $\psi_i$ is the order parameter. The coefficients $\alpha_i$ and $\beta_i$ denote the features that are temperature-dependent and temperature-independent, respectively, while $f_i(\hat{k})$ is the anisotropic function in which the subscripts $i$ ($i$ = 1, 2) represent the variables in the first and second band, the coefficients for the inter-band mixing of the two order parameters and their gradient are $\varepsilon$ and $\varepsilon_1$, respectively. The complex conjugate of the previous term appears as $c.c.$ in Equation (3), $\vec{A}$ is the vector potential; the series $\gamma_0 + \gamma_1 B + \gamma_2\frac{B^2}{2\mu_0}$ represents the magnetic field for a slight change in the $B$-field, where $B$ describes the magnetic field and $\gamma_0, \gamma_1$ and $\gamma_2$ are coefficient parameters [33,34].

$$\gamma_0 + \gamma_1 B + \gamma_2 \frac{B^2}{2\mu_0} = \int (B - \mu_0 M_{ions}) \frac{dB}{\mu_0} - (B - \mu_0 M) M_{sc} - (B - \mu_0 M) M_{ions} \quad (4)$$

where $B = \mu_0 H + \mu_0 M_{sc} + \mu_0 M_{ions}$ and $M_{ions} = \chi H_{c2}(T) + \chi'(H - M_{sc} - H_{c2})$, obtaining the magnetic field $B = \mu_0(\chi - \chi')H_{c2} + \mu_0(1 + \chi')(H + M_{sc})$, in which $\mu_0 H$, $\mu_0 M_{sc}$ and $\mu_0 M_{ions}$ represent the applied field, the field generated by carriers, and the field generated by ions, respectively, while $\chi$ is the susceptibility and $\chi'$ is the differential susceptibility.

When we applied a variation of Equation (1) (concerning $\psi_1^*$ and $\psi_2^*$), the 1st Ginzburg–Landau equation was found:

$$\frac{\langle f_1^2(\hat{k})\rangle}{2m_1}\left(-i\hbar\vec{\nabla} - 2e\vec{A}\right)^2\psi_1 + \alpha_1\left\langle f_1^2(\hat{k})\right\rangle\psi_1 + \varepsilon\left\langle f_1(\hat{k})f_2(\hat{k})\right\rangle\psi_2$$

$$+\varepsilon_1\left\langle f_1(\hat{k})f_2(\hat{k})\right\rangle\left(-i\hbar\vec{\nabla} - 2e\vec{A}\right)^2\psi_2 = 0 \quad (5)$$

$$\frac{\langle f_2^2(\hat{k})\rangle}{2m_2}\left(-i\hbar\vec{\nabla} - 2e\vec{A}\right)^2\psi_2 + \alpha_2\left\langle f_1^2(\hat{k})\right\rangle\psi_2 + \varepsilon\left\langle f_1(\hat{k})f_2(\hat{k})\right\rangle\psi_1$$

$$+\varepsilon_1\left\langle f_1(\hat{k})f_2(\hat{k})\right\rangle\left(-i\hbar\vec{\nabla} - 2e\vec{A}\right)^2\psi_1 = 0 \quad (6)$$

After substitution of $C = \frac{\psi_1}{\psi_2}$, $\psi_1(x) = e^{\frac{-\delta x^2}{2}}$ and $m_1 = m_2 = m$, the simplified equation was obtained as follows:

$$-\frac{\hbar^2}{2m}\left[C^2\left\langle f_1^2(\hat{k})\right\rangle - \left\langle f_2^2(\hat{k})\right\rangle\right]\frac{d^2\psi_1}{dx^2} + \frac{4e^2 A^2}{2m}\left[C^2\left\langle f_1^2(\hat{k})\right\rangle - \left\langle f_2^2(\hat{k})\right\rangle\right]\psi_1$$

$$+\left[\alpha_1 C^2\left\langle f_1^2(\hat{k})\right\rangle - \alpha_2\left\langle f_2^2(\hat{k})\right\rangle\right]\psi_1 = 0 \quad (7)$$

Using $\vec{A} = (0, \ B_0(x - x_0), \ 0)$ as the one-dimensional vector potential, which takes the magnetic field in the $z$-direction, and $\vec{B_0} = B_0\vec{z}$, $B_0 = \mu_0(\chi - \chi')H_{c2} + \mu_0(1 + \chi')(H + M_{sc})$, and identifying $\lambda = \frac{2e}{\hbar}$ and $\alpha_0 = -\frac{2e^2}{m}$, the equation can be rewritten as

$$\frac{d^2\psi_1}{dx^2} - \lambda^2 B_0^2(x - x_0)^2\psi_1 = \frac{\lambda^2}{\alpha_0}\frac{\left[\alpha_1 C^2\left\langle f_1^2(k)\right\rangle - \alpha_2\left\langle f_2^2(k)\right\rangle\right]}{\left[C^2\left\langle f_1^2(k)\right\rangle - \left\langle f_2^2(k)\right\rangle\right]}\psi_1 \quad (8)$$

By setting $\psi = \exp(-\frac{1}{2}b\xi^2)$, $\xi = (\lambda B_0)^{\frac{1}{2}}x$, $\xi_0 = (\lambda B_0)^{\frac{1}{2}}x_0$ and $\beta = \frac{\lambda^2}{\alpha_0}\frac{\left[\alpha_1 C^2\left\langle f_1^2(k)\right\rangle - \alpha_2\left\langle f_2^2(k)\right\rangle\right]}{\left[C^2\left\langle f_1^2(k)\right\rangle - \left\langle f_2^2(k)\right\rangle\right]}$, we can obtain the formula $-\frac{d^2\psi_1}{d\xi^2} + (\xi - \xi_0)^2\psi_1 = \beta\psi_1$. The variational problem of locating the minimum expression is equal to determining the minimum value of $\beta$ [4,30] $\beta = \frac{\int_0^\infty\left[\left(\frac{d\psi}{d\xi}\right)^2 + (\xi - \xi_0)\psi^2\right]d\xi}{\int_0^\infty\psi^2 d\xi} = \frac{b}{2} + \frac{1}{2b} - \frac{2\xi_0}{\sqrt{\pi b}} + \xi_0^2$. It was found that $\beta_{min} = b = \left(1 - \frac{2}{\pi}\right)^{\frac{1}{2}}$. Thus, the result is $B_0 = \frac{\lambda}{\left(1 - \frac{2}{\pi}\right)^{\frac{1}{2}}\alpha_0}\left[\frac{\alpha_1 C^2\left\langle f_1^2(\hat{k})\right\rangle - \alpha_2\left\langle f_2^2(\hat{k})\right\rangle}{C^2\left\langle f_1^2(\hat{k})\right\rangle - \left\langle f_2^2(\hat{k})\right\rangle}\right]$.

Because there is $B_0 = \mu_0(\chi - \chi')H_{c2} + \mu_0(1 + \chi')(H + M_{sc})$, the surface critical magnetic field depends on the anisotropic and temperature of the two-band magnetic superconductor as

$$H_{c3} = \frac{\frac{1.66}{\alpha_0}\left[\frac{\alpha_1(T)C^2\left\langle f_1^2(\hat{k})\right\rangle - \alpha_2(T)\left\langle f_2^2(\hat{k})\right\rangle}{C^2\left\langle f_1^2(\hat{k})\right\rangle - \left\langle f_2^2(\hat{k})\right\rangle}\right]H_{c2} - (\chi - \chi')H_{c2}}{(1 + \chi')} \quad (9)$$

where $H_{c3}$ is the surface critical magnetic field, $H_{c2}$ is the upper critical magnetic field, $\alpha_1$ and $\alpha_2$ are temperature-dependent functions in the first and second bands, respectively. $\alpha_0$ is a constant with the value $\left(\alpha_0 = -\frac{2e^2}{m}\right)$, $\chi$ is the susceptibility, $\chi'$ is the differential

susceptibility, $C$ corresponds to the quantity of energy band multiples, and $\left\langle f_1^2(\hat{k}) \right\rangle$ and $\left\langle f_2^2(\hat{k}) \right\rangle$ are anisotropic functions in the first and second bands, respectively.

To learn more about the temperature dependency of the surface critical magnetic field, there are four types of temperature-dependent functions [31], as follows: $\alpha_{Ch} = \left(1 - \frac{T}{T_c}\right)$ [35], $\alpha_{Zh} = \left(1 - \left(\frac{T}{T_c}\right)^2 / 1 + \left(\frac{T}{T_c}\right)^2\right)$ [20], $\alpha_{Sh} = \left(1 - \frac{T}{T_c}\right) + \frac{1}{2}\left(1 - \frac{T}{T_c}\right)^2$ [36] and $\alpha_{Ud} = p\left(1 - \frac{T}{T_c}\right) + \frac{q}{2}\left(1 - \frac{T}{T_c}\right)^2$ [37]. It was found that the temperature parameters are in the polynomial of second order, so the second-order quadratic form is set up as

$$\alpha_i = p_i\left(1 - \frac{T}{T_c}\right) + \frac{q_i}{2}\left(1 - \frac{T}{T_c}\right)^2, \ i = 1, 2 \tag{10}$$

The anisotropy is included in the consideration, and the anisotropic functions of ellipse and pancake forms [38,39] were used for $\left\langle f^2(\hat{k}) \right\rangle$. The wave vector was changed into an azimuthal angle then $\left\langle f^2(\hat{k}) \right\rangle \equiv \left\langle f^2(\theta) \right\rangle$. The anisotropic function consisting of an ellipse shape from the model by Hass and Maki was $f(\theta) = \frac{1 + a\cos^2\theta}{1 + a}$, where $a$ is the constant that affects anisotropy and $\theta$ is the azimuthal angle. The experimentally determined gap ratio results in $a \approx 1$ for the calculations provided in [38]. In momentum space, the anisotropic s-wave order parameter $\Delta(k) = \Delta(1 + z^2)/2$ is displayed. This function is an ellipsoid with a minor axis $\Delta/2$ in the a-b plane and a major axis $\Delta$ in the c direction. The smaller value of the a-b plane gap function is consistent with the larger in-plane Coulomb repulsion proposed in [40]. Likewise, the pancake shape of the model by Posazhennikova was $f(\theta) = \frac{1}{\sqrt{1 + b\cos^2\theta}}$, $b$ is the constant that affects anisotropy and $\theta$ is the azimuthal angle concerning the c-axis. For $b = 10$ [39], the anisotropic s-wave order parameter in momentum space has a pancake shape. Its maximum value $\Delta_{\min} = \Delta$ is in the ab-plane, while its minimum value $\Delta_{\min} = \Delta/\sqrt{1 + b}$ lies along the c-axis when $\Delta(k, T) \sim f(k)\Delta(T)$, demonstrating the link between the Ginzburg–Landau and the BCS theories $\psi \sim \Delta$ [41].

When setting $\left\langle f_1^2(\hat{k}) \right\rangle = 1$ and $\left\langle f_2^2(\hat{k}) \right\rangle = 1$, the surface critical magnetic field in Equation (9) can be reduced to an isotropic two-band magnetic superconductor, and it can be reduced to an isotropic one-band magnetic superconductor when setting $\alpha_1 = \alpha_0$ and $\alpha_2 = 0$ [30]. Likewise, for non-magnetic superconductors, $\chi = \chi' = 1$, it can be reduced to a well-known single-band superconductor formula [2].

After using Equation (10), Equation (9) can be simplified as

$$\frac{H_{c3}}{H_{c2}} = 1.66\left[k_1 + k_2\frac{T}{T_c} + k_3\left(\frac{T}{T_c}\right)^2\right] \tag{11}$$

In which

$$k_1 = \frac{p_1 C^2 - p_2\frac{\langle f_2^2(\hat{k})\rangle}{\langle f_1^2(\hat{k})\rangle} + \frac{q_1}{2}C^2 - \frac{q_2}{2}\frac{\langle f_2^2(\hat{k})\rangle}{\langle f_1^2(\hat{k})\rangle}}{\alpha_0(1 + \chi')\left(C^2 - \frac{\langle f_2^2(\hat{k})\rangle}{\langle f_1^2(\hat{k})\rangle}\right)} - \frac{(\chi - \chi')}{1 + \chi'},$$

$$k_2 = \frac{-p_1 C^2 + p_2\frac{\langle f_2^2(\hat{k})\rangle}{\langle f_1^2(\hat{k})\rangle} - q_1 C^2 + q_2\frac{\langle f_2^2(\hat{k})\rangle}{\langle f_1^2(\hat{k})\rangle}}{\alpha_0(1 + \chi')\left(C^2 - \frac{\langle f_2^2(\hat{k})\rangle}{\langle f_1^2(\hat{k})\rangle}\right)},$$

$$k_3 = \left[\frac{q_1}{2}C^2 - \frac{q_2}{2}\frac{\langle f_2^2(\hat{k})\rangle}{\langle f_1^2(\hat{k})\rangle}\right]\frac{1}{\alpha_0(1 + \chi')\left(C^2 - \frac{\langle f_2^2(\hat{k})\rangle}{\langle f_1^2(\hat{k})\rangle}\right)}. \tag{12}$$

There is a singularity point at $C^2 = \frac{\langle f_2^2(\hat{k})\rangle}{\langle f_1^2(\hat{k})\rangle}$ or $C = \pm\sqrt{\frac{\langle f_2^2(\hat{k})\rangle}{\langle f_1^2(\hat{k})\rangle}}$ where the wave function and the average of the anisotropic function are coincidental. The crucial parameters are $k_1$, $k_2$ and $k_3$, while $p_i$ and $q_i$ are subparameters that denote arbitrary constant of the temperature-dependent function of $\alpha_{Ud}$, and the subscripts $i = 1, 2$ specify the variables in the first and second energy bands. Because there is the upper critical magnetic field in Equation (11), the same process was applied to find the temperature dependence of $H_{c2}(T)$ [37].

To calculate the upper critical magnetic field, one can substitute the vector potential $\vec{A} = [\mu_0(\chi - \chi')H_{c2}x + \mu_0(1+\chi')(H + M_{sc})x]\hat{j}$ into the 1st Ginzburg–Landau equation in Equations (5) and (6), set the magnetic parameter $l_s^2 = \frac{\hbar^2}{4e^2(\mu_0(\chi-\chi')H_{c2}x + \mu_0(1+\chi')(H+M_{sc})x)}$ (where $\mu_0$ represents the magnetic permeability), and assign the constant values $\lambda_1$, $\lambda_1$, $a$ and $b$ to the wave function $\psi_1 = \lambda_1 e^{\frac{-ax^2}{2}}$ and $\psi_2 = \lambda_2 e^{\frac{-bx^2}{2}}$ [32,42,43], to obtain

$$-\frac{\hbar^2}{2m_1}\left\langle f_1^2(\hat{k})\right\rangle\left[-a(1 - ax^2) - \frac{x^2}{l_s^2}\right]\psi_1 + \alpha_1\left\langle f_1^2(\hat{k})\right\rangle\psi_1 + \varepsilon\left\langle f_1(\hat{k})f_2(\hat{k})\right\rangle\psi_2$$
$$+\hbar^2\varepsilon_1\left\langle f_1(\hat{k})f_2(\hat{k})\right\rangle\left[-b(1 - bx^2) - \frac{x^2}{l_s^2}\right]\psi_2 = 0 \tag{13}$$

$$-\frac{\hbar^2}{2m_2}\left\langle f_2^2(\hat{k})\right\rangle\left[-b(1 - bx^2) - \frac{x^2}{l_s^2}\right]\psi_2 + \alpha_2\left\langle f_2^2(\hat{k})\right\rangle\psi_2 + \varepsilon\left\langle f_1(\hat{k})f_2(\hat{k})\right\rangle\psi_1$$
$$+\hbar^2\varepsilon_1\left\langle f_1(\hat{k})f_2(\hat{k})\right\rangle\left[-a(1 - ax^2) - \frac{x^2}{l_s^2}\right]\psi_1 = 0 \tag{14}$$

When Equations (13) and (14) are written in matrix form, the result is $a^2 = \frac{1}{l_s^2}$ and $b^2 = \frac{1}{l_s^2}$ when $\frac{\hbar^2}{2m_1 l_s^2}\left\langle f_1^2(\hat{k})\right\rangle - \frac{\hbar^2 a^2}{2m_1}\left\langle f_1^2(\hat{k})\right\rangle = 0$ and $\frac{\hbar^2}{2m_2 l_s^2}\left\langle f_2^2(\hat{k})\right\rangle - \frac{\hbar^2 b^2}{2m_2}\left\langle f_2^2(\hat{k})\right\rangle = 0$, respectively, and set $m_1 = m_2 = m$, causing a temperature-dependent upper critical magnetic field $H_{c2}(T)$, as shown in the equation below:

$$H_{c2} = \frac{-m(\alpha_1 + \alpha_2 + 4\varepsilon\varepsilon_1\Omega m)}{\hbar e(1+\chi)(1 - 4\varepsilon_1^2\Omega m^2)} + \frac{m(\alpha_1\alpha_2 + \Omega\varepsilon^2)}{\hbar e(1+\chi)(\alpha_1 + \alpha_2 + 4\varepsilon\varepsilon_1\Omega m)} \tag{15}$$

Using the temperature dependence as $\alpha_i(T) = p_i(1 - \frac{T}{T_c}) + \frac{q_i}{2}(1 - \frac{T}{T_c})^2$ and $\alpha_{i0}(T) = p_{i0} + \frac{q_{i0}}{2}$, $i = 1, 2$, the result is

$$H_{c2} \approx H_{c2}^0 + R_1\frac{T}{T_c} + R_2\left(\frac{T}{T_c}\right)^2 \tag{16}$$

In which

$$H_{c2}^0 = \frac{-m(\alpha_{10} + \alpha_{20} + 4\varepsilon\varepsilon_1\Omega m)}{\hbar e(1+\chi)(1 - 4\varepsilon_1^2\Omega m^2)} + \frac{m(\alpha_{10}\alpha_{20} - \Omega\varepsilon^2)}{\hbar e(1+\chi)(\alpha_{10} + \alpha_{20} + 4\varepsilon\varepsilon_1\Omega m)},$$

$$R_1 = \frac{m(p_1 + p_2 + q_1 + q_2)}{\hbar e(1+\chi)(1 - 4\varepsilon_1^2\Omega m^2)} - \frac{m(2p_1p_2 + \frac{3p_1q_2}{2} + \frac{3p_2q_1}{2} + q_1q_2)}{\hbar e(1+\chi)(\alpha_{10} + \alpha_{20} + 4\varepsilon\varepsilon_1\Omega m)},$$

$$R_2 = \frac{-m(\frac{q_1}{2} + \frac{q_2}{2})}{\hbar e(1+\chi)(1 - 4\varepsilon_1^2\Omega m^2)} + \frac{m(p_1p_2 + \frac{3p_1q_2}{2} + \frac{3p_2q_1}{2} + \frac{q_1q_2}{2})}{\hbar e(1+\chi)(\alpha_{10} + \alpha_{20} + 4\varepsilon\varepsilon_1\Omega m)}. \tag{17}$$

where $H_{c2}^0$ is the zero-temperature upper critical magnetic field and $\Omega = \frac{\langle f_1(\hat{k})f_2(\hat{k})\rangle^2}{\langle f_1^2(\hat{k})\rangle\langle f_2^2(\hat{k})\rangle}$, <.....> is the average over Fermi surface. $H_{c2}^0$, $R_1$, and $R_2$ are the crucial variables, whereas $e$ is electron charge, $m$ is electron mass and $\varepsilon$ and $\varepsilon_1$ are inter-band mixing of the two order parameters and their gradient.

## 3. Results and Discussion

The simplified equation for the temperature-dependent surface and upper critical magnetic field of anisotropic two-band superconductors is in the second-order quadratic form as

$$\frac{H_{c3}(T)}{H_{c2}(T)} = 1.66\left[k_1 + k_2\frac{T}{T_c} + k_3\left(\frac{T}{T_c}\right)^2\right]$$

$$H_{c2}(T) \approx H_{c2}^0 + R_1\frac{T}{T_c} + R_2\left(\frac{T}{T_c}\right)^2$$

Here, the constants $k_1, k_2, k_3, R_1, R_2$ and $H_{c2}^0$ can be fit by using the experimental data that can give information on the temperature-dependent behavior and anisotropic type of superconductor.

FeSe refers to a distinct class of binary iron–chalcogenide materials (FeCh; Ch = chalcogens) characterized as "without charge reservoir layers" [28,44]. In general, structural parameters change for $AFe_2Se_2$ phases that can be accomplished through cation substitutions (in A- or (and) iron sites) or anion substitutions in Se sites. Data show that changes in A sites have the greatest influence on parameter c during the series [45]; the c parameters rise while parameter a fluctuates very little. For example, replace A with K as $KFe_2Se_2$, $RbFe_2Se_2$, and $CsFe_2Se_2$. Changes in the anion sites could cause significant changes in the electronic properties of ternary iron-chalcogenide superconductors, and the structure of these substances matches their bandgap and Fermi surface topology. The fundamental properties of electronic bands and the Fermi surface topology result in near-fermi bands having a complex "mixed" nature that is strongly anisotropic. The Fermi level is crossed by quasiflat and high-dispersive bands; the Fermi surface is completely electronic-like, and the bandwidth narrows as the particle densities of states increase. However, there is little experimental data on the surface and upper critical magnetic fields of the superconductors that only have for $K_{0.73}Fe_{1.68}Se_2$ and $La_{1.85}Sr_{0.15}CuO_4$ superconductors. The $K_{0.73}Fe_{1.68}Se_2$ superconductor is one of the superconductors in the $MFe_2Se_2$ type (M = K, Rb, Cs, Tl/K and Tl/Rb), which exhibit long-range three-dimensional antiferromagnetism at $T_N \approx 140$–150 K and 520–550 K [22]. The $Fe^{2+}$ moments are along the c-axis [46]. The structure and magnetism transition are tetragonal to orthorhombic for the preceding pnictides, but there is a tetragonal (I4/mmm) structure to another tetragonal (I4/m) structure for Fe-Se based. The nonstoichiometric $K_xFe_{2-x}Se_2$ materials become superconductors at about 30–33 K, during which the antiferromagnetic state persists even at a low temperature. Thus, the superconducting state and antiferromagnetic state are coexistent in this material. Both states are in the same Fe-Se crystallographic layer [47]. The study of this material has discovered a new interplay between magnetism and superconductivity, which should usher in a new era of superconducting and magnetism coexistence. The study on $K_{0.73}Fe_{1.68}Se_2$ superconductor [22] discovered that there was no symmetry hysteresis, that AFM and Fm particles coexisted, and that the diamagnetic signal from the Fe effect was found.

To apply the formula in this study to the $K_{0.73}Fe_{1.68}Se_2$ superconductor, the experimental data $H_{c2}$ and $H_{c3}$ were fitted with a second-order polynomial, and the coefficients were read out and solved. Using the constants in Equations (18) and (19) and their substitution into Equations (12) and (17) and solving for the internal parameters of superconductors, the equation yield $C = 2$, $\alpha_0 = 0.001$, $\chi = 1$, $\chi' = -0.72$, $\frac{m}{\hbar e} = 300$, $\varepsilon_1 m = 0.1$, $\varepsilon = 1$, $p_1 = -0.888285$, $p_2 = -3.89599$, $q_1 = 1$ and $q_2 = 7.79991$.

The constants were found to be

$$\frac{H_{c3}(T)}{H_{c2}(T)} = 1.66\left[-1092.95 + 2285.32\frac{T}{T_{c2}} - 1188.66\left(\frac{T}{T_c}\right)^2\right] \qquad (18)$$

$$H_{c2}(T) \approx 416.37 - 618.59\frac{T}{T_c} + 202.35\left(\frac{T}{T_c}\right)^2 \qquad (19)$$

Equations (18) and (19) were used to calculate the curve with $T_c = 30.8$ K, and the behavior of $\frac{H_{c3}(T)}{H_{c2}(T)}$ and $H_{c2}(T)$ versus temperature is shown in Figure 1. Here, the upper critical magnetic field decreases as the temperature increases and reaches zero at a critical temperature. The zero-temperature upper critical of 416.37 kOe was found to be in the range from 193 kOe to 60 T of the rough estimation for Ref. [22] and well extrapolated in a wide range of temperatures for Ref. [48].

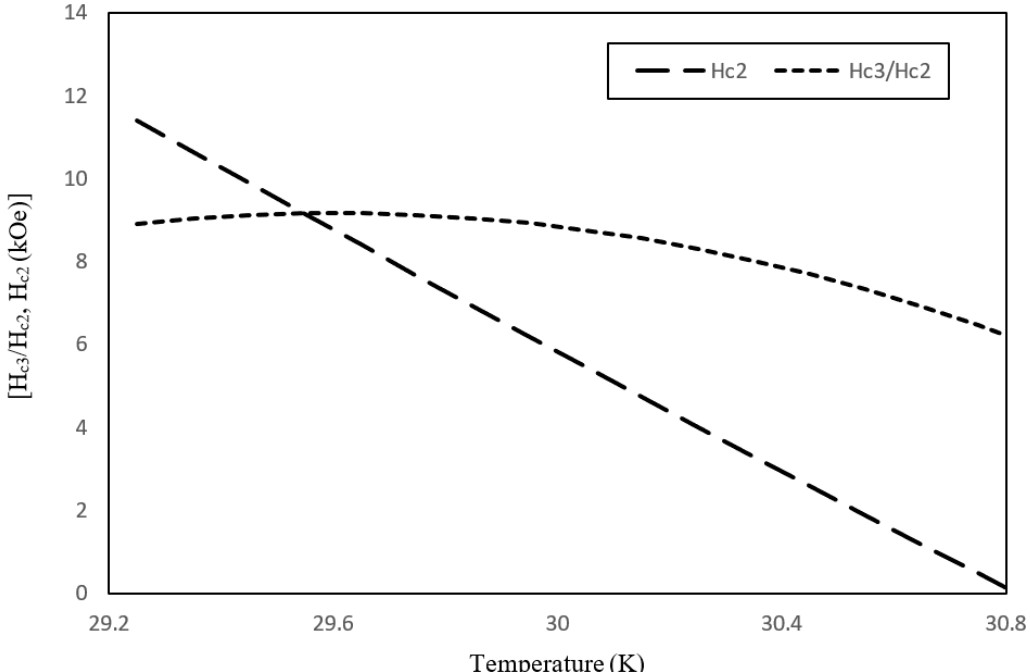

**Figure 1.** The $\frac{H_{c3}(T)}{H_{c2}(T)}$ and $H_{c2}(T)$ versus temperature ($T$) of $K_{0.73}Fe_{1.68}Se_2$ superconductor.

The calculation in this study found that $\frac{H_{c3}(T_c)}{H_{c2}(T_c)} \approx 6.2$ was higher than that reported by $\frac{H_{c3}(T_c)}{H_{c2}(T_c)} \approx 4.4$ [22]. However, the result showed the temperature dependence on $\frac{H_{c3}(T)}{H_{c2}(T)}$ value was higher than 1.66 for a well-known formula. As the temperature increases, $\frac{H_{c3}(T)}{H_{c2}(T)}$ is decreased to the lowest value of 6.2. It was found that the magnetic parameters such as $\chi$ and $\chi'$ showed a higher effect on $\frac{H_{c3}(T_c)}{H_{c2}(T_c)}$ than the other parameters at the critical temperature.

The anisotropic behavior is appropriate for a pancake shape: $\left( f(\theta) = \frac{1}{\sqrt{1 + a_i \cos^2 \theta}}, i = 1, 2 \right)$ where $\frac{\langle f_2^2(\hat{k}) \rangle}{\langle f_1^2(\hat{k}) \rangle} = 0.94555$ and $\Omega = \frac{\langle f_1(\hat{k}) f_2(\hat{k}) \rangle^2}{\langle f_1^2(\hat{k}) \rangle \langle f_2^2(\hat{k}) \rangle} = 0.5$, with $a_1 = 2.49785$ and $a_2 = 3.39198$. The Fermi surfaces of each band were plotted so that they were nearly identical in shape, as shown in Figure 2.

This study demonstrated the temperature and anisotropy dependence of the pancake-pancake shape in the band structure of the $K_{0.73}Fe_{1.68}Se_2$ superconductor. The anisotropic characteristics of the $K_{0.73}Fe_{1.68}Se_2$ superconductor have been studied and shown to be anisotropic in the upper and surface critical magnetic fields [22,48]. Here, the two-band model was utilized for the computation. However, the findings were nearly the same for both bands. Our findings are consistent with the experimental data, which reveal that the geometry of the upper critical magnetic field has a high value in the same range [22,46–48].

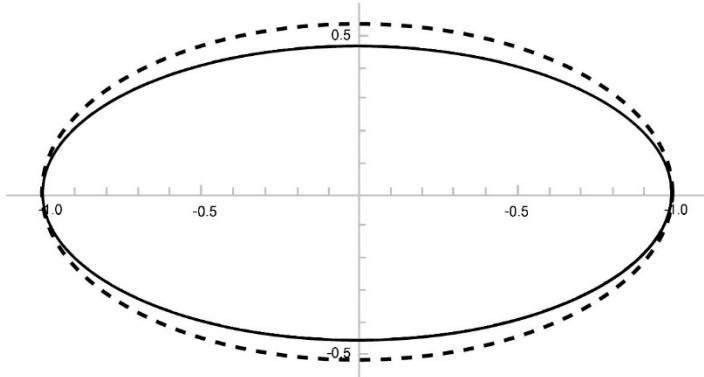

**Figure 2.** The anisotropic shape of the first and second band of $K_{0.73}Fe_{1.68}Se_2$ superconductor.

Superconducting cuprate: LaSrCuO (LSCO) changes into a normal conductor in the presence of a strong magnetic field and a low temperature [49]. In the crystal lattice, ions of the rear earth elements divide layers of copper oxide planes [50]; the spacing between these ions is approximately 3.78 Angstrom. There are two layers of La (Sr)-O planes between neighboring Cu-O planes in LSCO. Doping alters the charge carrier concentration in LSCO superconductors. The valance is $La^{3+}$ and $O^{2+}$. Therefore, all the Cu ions in the parent molecule ($x = 0$) are in the $Cu^{2+}$ state, i.e., they contain one unpaired electron in a d shell. As $x$ goes up, the number of carriers on the Cu-O planes is controlled by "charge reservoirs" placed between the planes. Because the valence of Sr is only 2+, increasing its Sr concentration to $x$ pulls the electrons from the copper planes and causes holes in the copper sites. The concentration of holes in LSCO is related to the Sr of the unit cell. As doping fluctuates, the physical characteristics of cuprates alter dramatically. As the doping $x$ and temperature $T$ are varied, several phases with unusual physical features have been identified [51]. On the Cu-O planes of the undoped parent material, the electron spins are organized in an antiferromagnetic (AF) configuration. Once holes are added to a system, the antiferromagnetic order across long distances is destroyed. At $x = 0$, the $T_N$ of this phase is close to room temperature, and it drops quickly with small changes in $x$ until it is gone at $x = 0.02$. In contrast, high $T_c$ superconductivity ranges from $x = 0.05$ to $x = 0.26$, with the greatest transition temperature $T_c$ of about 38 K occurring at $x = 0.15$. Numerous tests indicate that the order parameter of cuprates possesses d-wave symmetry, i.e., $\Delta(k) = \Delta_0(\cos(k_x) - \cos(k_y))$. In the area between AF and SC, $x = 0.02$ and $x = 0.05$, a spin glass phase (SG) containing short-range magnetic order coexists with superconductivity up to $x = 0.08$ [52]. Above $x = 0.27$, superconductivity disappears and LSCO acts like a normal metal. Research was carried out on $La_{1.85}Sr_{0.15}CuO_4$ single crystals using tiny magnetic fields. This enabled the measurement of the AB-plane and C-axis diamagnetic responses. The primary findings of this study were anisotropic between the two directions of the superconducting transition temperature. All samples positioned perpendicular to the planes demonstrated consistently greater $T_c$ than those oriented parallel to the planes.

$La_{1.85}Sr_{0.15}CuO_4$ is another superconductor that has sufficient data for consideration. $La_{1.85}Sr_{0.15}CuO_4$ single crystals [9] have been measured for the temperature dependence of the upper and surface critical fields. The paramagnetic properties were found and shown to exhibit the paramagnetic Meissner effect under critical temperatures. The $H_{c2}(0)$ was estimated with $H_{c2}^a(0) = 11.6$ T and $H_{c2}^c(0) = 4.1$ T for a- and c-crystals, respectively. The surface critical field was estimated linearly near $T_c$, resulting in an anisotropy effect with $\frac{H_{c3}}{H_{c2}} = 1.80$ and 4.0 in c- and a-crystals near the critical temperature (35 K).

The experimental data for the $La_{1.85}Sr_{0.15}CuO_4$ superconductor in the a-direction, LaSrCuO-a, were fitted using a second-order polynomial, and the coefficients were read out and solved using Equations (12) and (17). The parameters include $C = 2$, $\alpha_0 = 0.1$,

$\chi = -0.2$, $\chi' = 0.1$, $\frac{m}{\hbar e} = 10$, $\varepsilon_1 m = 0.1$, $\varepsilon = 1$, $p_1 = -1.0951$, $p_2 = 1.0583$, $q_1 = -10$ and $q_2 = -0.980902$, with $T_c = 34.8$ K.

Giving

$$\frac{H_{c3}(T)}{H_{c2}(T)} = 1.66 \left[ 3.69 - 1.12 \frac{T}{T_{c\,2}} - 2.41 \left( \frac{T}{T_c} \right)^2 \right] \tag{20}$$

$$H_{c2}(T) \approx 84.728 - 156.42 \frac{T}{T_c} + 71.694 \left( \frac{T}{T_c} \right)^2 \tag{21}$$

The anisotropic behavior is suitable for a pancake shape $\left( f(\theta) = \frac{1}{\sqrt{1 + a_i \cos^2 \theta}}, i = 1, 2 \right)$ in which $\frac{\langle f_2^2(\hat{k}) \rangle}{\langle f_1^2(\hat{k}) \rangle} = 357.941$ and $\Omega = \frac{\langle f_1(\hat{k}) f_2(\hat{k}) \rangle^2}{\langle f_1^2(\hat{k}) \rangle \langle f_2^2(\hat{k}) \rangle} = 0.5$ with $a_1 = 919525$ and $a_2 = 3.33332$.

Figure 3 shows the $\frac{H_{c3}^a(T)}{H_{c2}^a(T)}$ and $H_{c2}^a(T)$ versus the temperature of the LaSrCuO-a supercon-ductor calculated using Equations (20) and (21); the $\frac{H_{c3}^a(T_c)}{H_{c2}^a(T_c)} \approx 0.3$ and $H_{c2}^a(T = 0) = 84.728$ T $T_c = 34.8$ K were found. Figure 4 shows the anisotropic shape of the first and second bands, with flat band behavior in the first band.

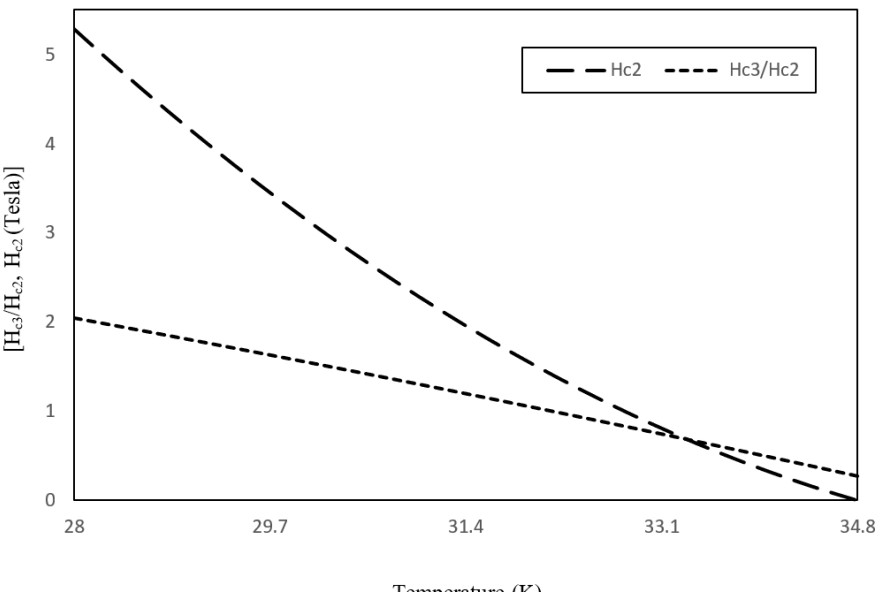

**Figure 3.** The $\frac{H_{c3}^a(T)}{H_{c2}^a(T)}$ and $H_{c2}^a(T)$ versus temperature of the La$_{1.85}$Sr$_{0.15}$CuO$_4$ -a superconductor.

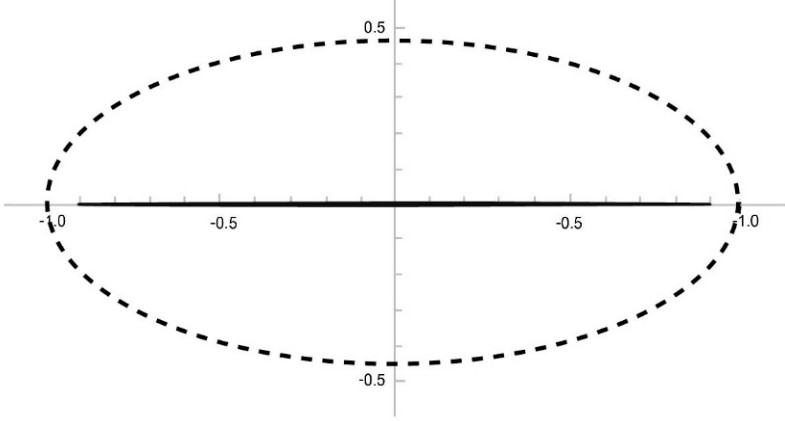

**Figure 4.** The anisotropic shape of the first and second bands of La$_{1.85}$Sr$_{0.15}$CuO$_4$ -a superconductor.

The experimental data for the $La_{1.85}Sr_{0.15}CuO_4$ superconductor in the c-direction, LaSrCuO-c, were fitted using a second-order polynomial, and the coefficients were read out and solved using Equations (12) and (17). The parameters included $C = 2$, $\alpha_0 = 0.1$, $\chi = 1$, $\chi' = -4.2$, $\frac{m}{\hbar e} = 10$, $\varepsilon_1 m = 0.1$, $\varepsilon = 1$, $p_1 = 2.38547$, $p_2 = -1.827$, $q_1 = -10$ and $q_2 = 1.48924$, with $T_c = 34.8$ K.

Thus,

$$\frac{H_{c3}(T)}{H_{c2}(T)} = 1.66\left[2.598 + 1.352\frac{T}{T_{c2}} - 2.991\left(\frac{T}{T_c}\right)^2\right] \tag{22}$$

$$H_{c2}(T) \approx 14.446 - 27.526\frac{T}{T_c} + 13.079\left(\frac{T}{T_c}\right)^2 \tag{23}$$

The anisotropic behavior is suitable for a pancake shape $\left(f(\theta) = \frac{1}{\sqrt{1+a_i\cos^2\theta}}, i = 1,2\right)$ in which $\frac{\langle f_2^2(\hat{k})\rangle}{\langle f_1^2(\hat{k})\rangle} = 31.5626$ and $\Omega = \frac{\langle f_1(\hat{k})f_2(\hat{k})\rangle^2}{\langle f_1^2(\hat{k})\rangle\langle f_2^2(\hat{k})\rangle} = 0.5$ with $a_1 = 7050.9$ and $a_2 = 3.33228$.

Figure 5 shows the $\frac{H_{c3}^c(T)}{H_{c2}^c(T)}$ and $H_{c3}^c(T)$ versus temperature of the LaSrCuO-c superconductor calculated using Equations (22) and (23); the $\frac{H_{c3}^c(T_c)}{H_{c2}^c(T_c)} \approx 1.6$ and $H_{c2}^c(T = 0) = 14.446$ T $T_c = 34.8$ K were found. Figure 6 shows an anisotropic shape with one flat band in the c-direction.

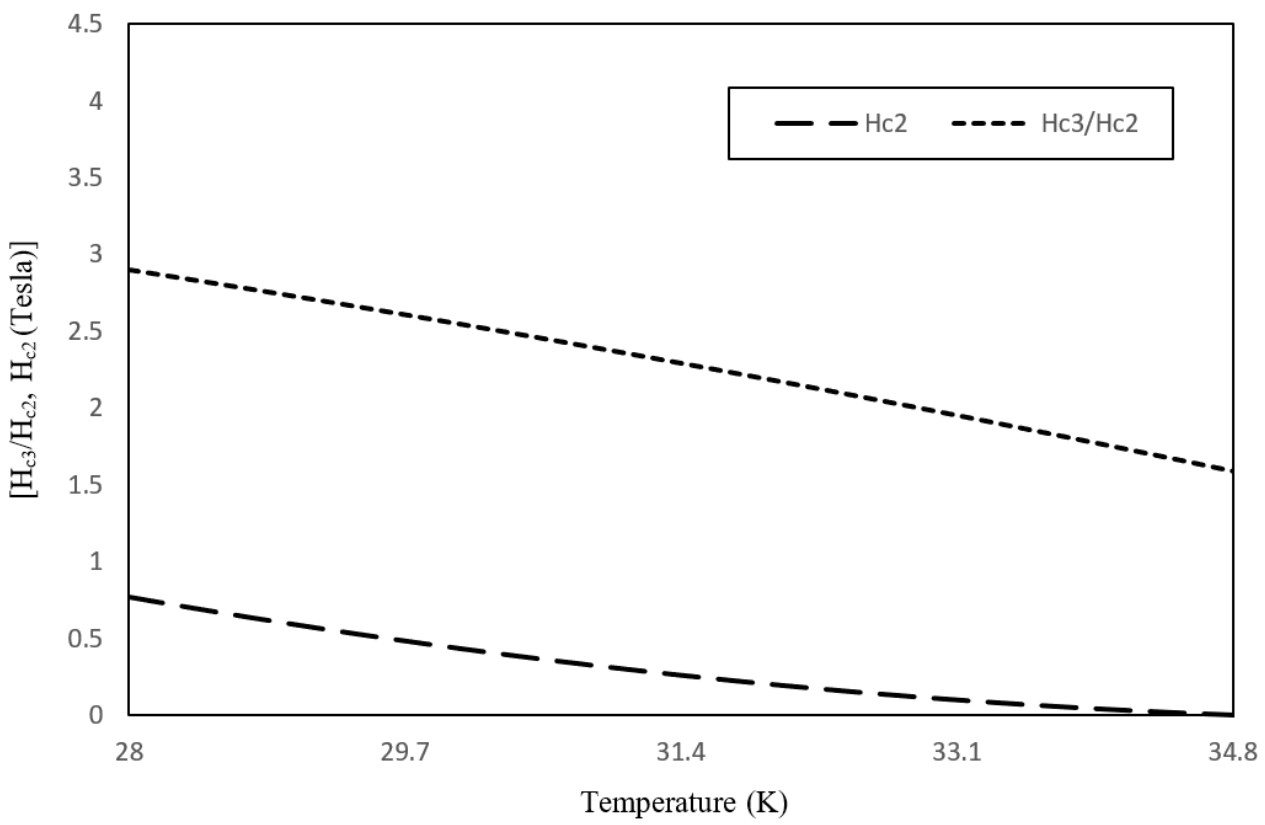

**Figure 5.** The $\frac{H_{c3}^c(T)}{H_{c2}^c(T)}$ and $H_{c2}^c(T)$ versus temperature of the $La_{1.85}Sr_{0.15}CuO_4$ -c superconductor.

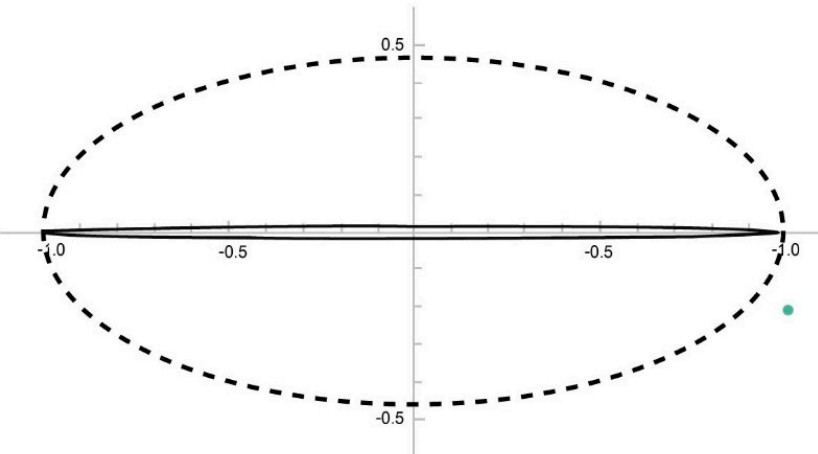

**Figure 6.** The anisotropic shape of the first and second bands of La$_{1.85}$Sr$_{0.15}$CuO$_4$ -c superconductor.

It was found that $H_{c2}^a(0) = 84.728$ T and $H_{c2}^c(0) = 14.446$ T were in the range of $H_{c2}^a(0) = 75$ T [53]. The highly anisotropic properties of the La$_{1.85}$Sr$_{0.15}$CuO$_4$ superconductor were discovered. The findings are consistent with a large magnitude difference in the c- and a- directions [9]. In both directions, the $\frac{H_{c3}(T)}{H_{c2}(T)}$ ratio decreases as the temperature increases. The magnetic parameters $\chi$ and $\chi'$ have a greater impact on the $\frac{H_{c3}(T)}{H_{c2}(T)}$ ratio at critical temperatures than the other. According to the report in Ref. [9], $\frac{H_{c3}^a(T_c)}{H_{c2}^a(T_c)} \approx 4.2$ and $\frac{H_{c3}^c(T_c)}{H_{c2}^c(T_c)} \approx 1.8$ with $T_c \approx 35$ K in an experiment that was conducted using linear extrapolation to estimate the value at critical temperatures. By using the two-band model, it could be found that $\frac{H_{c3}^a(T_c)}{H_{c2}^a(T_c)} \approx 0.3$ and $\frac{H_{c3}^c(T_c)}{H_{c2}^c(T_c)} \approx 1.6$ were in the same range of convention value at 1.66. The influence of magnetic parameters on the $\frac{H_{c3}(T_c)}{H_{c2}(T_c)}$ ratio that was incorporated in the computation is significant.

The upper critical magnetic field of LaSrCuO was determined using the London penetration depth and coherence length methods in order to predict the upper critical field $H_{c2}$ ($T$) for zero-temperature in superconductors with weak electron–phonon coupling limit, as described in Ref. [53]. Additionally, the Werthamar–Helfand–Honenberg method was employed as outlined in Ref. [9]. Both of these models utilize experimental data in order to describe the upper critical magnetic field of cuprate superconductors. The temperature-dependent upper critical magnetic field was analyzed by Talantsev et al. [54] and the modified Gorter–Casimir model [55–58] and modified WHH [59–64] were used to fit the experimental data with the degree 4th polynomial [54]. The parameters used are zero-temperature coherence length, zero-temperature energy gap, zero-temperature penetration depth, specific heat jump and critical temperature under the BCS approximation. Our model attempts to describe both the upper and surface critical magnetic fields. Specifically, we assume temperature-dependent parameters for a second-order polynomial, with free energy calculated within the region of the GL theory, to estimate the upper critical fields $H_{c2}$ ($T$) and $H_{c3}$ ($T$) near zero temperature. After fitting experimental data, the temperature relation of our model can be used to represent the zero temperature of the upper and surface critical magnetic fields. It should be interesting to analyze the experimental data of upper and surface critical magnetic fields using the various model; this requires raw data, which cover the reduced temperature, to estimate the zero-temperature critical magnetic field that is not available. We found that the zero-temperature upper critical magnetic fields for KFeSe, LaSrCuO-a, and LaSrCuO-c are 416.37 kOe, 84.728 T, and 14.446 T, respectively.

## 4. Conclusions

To sum up, this work investigated the surface critical magnetic field ($H_{c3}$) and the second critical magnetic field ($H_{c2}$) by taking into account the temperature-dependent function and the energy gap anisotropic function. Beginning with the Ginzburg–Landau equation for free energy, two bands of magnetic superconductors were identified. By minimizing the free energy of two bands $\left( \frac{\partial F_{sc}}{\partial \psi_1^*}, \frac{\partial F_{sc}}{\partial \psi_2^*} \right)$, two equations were obtained for the first Ginzburg–Landau equation. The first Ginzburg-Landau equation was simplified using the variation method to acquire the critical magnetic field at the surface. Then, the temperature-dependent functions were implemented. After that, the $H_{c3}$ was expressed in second-order polynomial form. The $H_{c2}$ was also reduced to its second-order polynomial by applying the temperature-dependent expressions, and the ratio for $H_{c3}/H_{c2}$ in terms of $T/T_c$ was determined. To assess the outcomes, the values of six crucial parameters were analyzed using the iteration method: $k_1$, $k_2$, $k_3$, $H_{c2}^0$, $R_1$ and $R_2$. The energy gap anisotropy function was shaped like a pancake–pancake by these important parameters; both the first and second bands had the shape of a pancake. The results were fitted to the experiments with iron-based superconductors (KFeSe) and cuprate superconductors (LaSrCuO). The $T_c$ of $K_{0.73}Fe_{1.68}Se_2$ was 30.8 K. As the temperature increased, both $H_{c2}$ and $H_{c3}/H_{c2}$ decreased, with $H_{c2}$ reaching zero at $T_c$ and $H_{c3}/H_{c2}$ achieving roughly 6.2 at $T_c$. This is consistent with the experimental results [22,48], where $H_{c3}/H_{c2}$ is greater than the experimental result [22] and $H_{c2}$ at 0 K is within the experimental range [48]. Moreover, the shapes of the Fermi surfaces in the pancake shapes of the two bands were essentially similar. $La_{1.85}Sr_{0.15}CuO_4$ had $T_c$ of 34.8 K. $H_{c2}$ and $H_{c3}/H_{c2}$ fell as temperature increased along the a- and c-directions, with $H_{c2}^a$ and $H_{c2}^c$ reaching zero at $T_c$ and $H_{c3}^a/H_{c2}^a$ and $H_{c3}^c/H_{c2}^c$ approaching around 0.3 and 1.6, respectively. This fits with the experimental results [9,53], where $H_{c3}/H_{c2}$ are lower than the experimental results [9], but within the range of the conventional value [2], and $H_{c2}$ at 0 K is within the experimental range [53]. In addition, the pancake shapes of the Fermi surface in the first band exhibited flat band behavior. Thus, $La_{1.85}Sr_{0.15}CuO_4$ is a superconductor with a single band. The temperature-dependent $H_{c3}/H_{c2}$ ratios of $K_{0.73}Fe_{1.68}Se_2$ and $La_{1.85}Sr_{0.15}CuO_4$ were higher than and within the same range as the well-known formula [2], respectively, which, in the calculation for this study, shows that the influence of magnetic parameters ($\chi$ and $\chi'$) has an effect on the $H_{c3}/H_{c2}$ ratio. The discovery of the shape of the Fermi surface reveals the microscopic structure of Fermi energy, which is significantly beneficial to researchers.

**Author Contributions:** S.M. calculates the surface and upper critical magnetic fields, analyzes the result, compares it to experimental data, and writes a manuscript. P.U. and A.C. developed the surface and upper critical magnetic field calculations model and validated all study findings. G.C. and T.K. examine the equations and revise the manuscript. All authors have read and agreed to the published version of the manuscript.

**Funding:** This research received financial support from the Royal Golden Jubilee Ph.D. Programme (Grant No. PHD/0212/2561) through the National Research Council of Thailand (NRCT) and the Thailand Research Fund (TRF).

**Data Availability Statement:** The data sets that support the findings in this study are available from the corresponding author upon reasonable request.

**Acknowledgments:** Our wish is to acknowledge the encouragement from Prasarnmit Physics Research Unit, Department of Physics, Faculty of Science, Srinakharinwirot University, and the financial support from the Royal Golden Jubilee Ph.D. Programme (Grant No. PHD/0212/2561) through the National Research Council of Thailand (NRCT) and Thailand Research Fund (TRF).

**Conflicts of Interest:** The authors declare no competing financial interest.

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
