# Peer review of "A Study of the Temperature-Dependent Surface and Upper Critical Magnetic Fields in KFeSe and LaSrCuO Superconductors"

_crystals, doi:10.3390/cryst13030526_

Round 1
Reviewer 1 Report
Presented study aimed to investigate the surface and upper critical magnetic fields of Fe-based superconductors . The anisotropic two-band Ginzburg-Landau method was used for analytic equation. A second-order polynomial temperature-dependent equation could be applied and fitted to the experimental results of new compounds KFeSe and LaSrCuO. Numerical calculations were applied to find the shape of the Fermi surface, which is an important component within the band structure. It is shown that anisotropy of the Fermi surface for each band structure was affected by the upper critical magnetic field Hc2 and the surface critical magnetic field Hc3.
Obtained results is soundable and deserve publication.
Before publication should be removed tecnhical incorrectness in text.
Reviewer 2 Report
referee report
crystals-2277272-peer-review-v1
A study of the temperature-dependent surface and upper critical magnetic fields in KFeSe and LaSrCuO superconductors
Suppanyou Meakniti et al.
This manuscript deals with a theoretical calculation concerning the upper critical field,
Hc2, and the surface critical field, Hc3. The obtained equations were applied and fitted
to experimental data of two quite special superconductors, KFeSe and LaSrCuO. The topic
is surely interesting and may fit to Crystals.
The present manuscript has 6 figures, no table, and 40 references are given.
The manuscript is overall well written with many equations, and several equations, e.g.,
9, 11, 13, 14 contain some symbols which do not belong there. This should be checked by
the authors carefully.
To fit better to the scope of Crystals, the authors should include the respective crystal
lattices of the materials under investigation, and it is also very important that the
proper chemical formulae are given. The best place for this would be in the introduction,
followed by a proper table in Section 2.
Another point of criticism are the figures -- the authors selected only a norrow range of
temperatures around Tc, which may be informative, but is certainly not the range where such
materials may be applied. Furthermore, it is also quite strange that the authors do not
attempt a discussion of their equations with the ones commonly employed in this field like the WHH model and others. There are many examples in the literature to compare with the new calculations.
Thus, the present manuscript requires major revisions.
Round 2
Reviewer 1 Report
no comments
Author Response
We didn't include the edit file as there were no comments provided by the reviewer for us to edit
Reviewer 2 Report
I appreciate the changes made by the authors, which clearly improve the manuscript. Of course, the addition of a respective drawing of the crystal structures would have been better, but this is decision of the authors.
Concerning the WHH model and others, the changes made are o.k., but there could have been much more information. See,e.g., the work by Talantsev et al., Sci. Rep. 10, 212, 2020. This is on a totally different superconducting material, but all equations and calculations there could be used for a better comparison of the present calculation and the existing approaches. Furthermore, for experimentalists, the value of Hc2(0K) is very important, which also should be mentioned in the present paper (and be compared to the other predictions).
Thus, I still recommend some more intense revision of the manuscript.
Round 3
Reviewer 2 Report
Well performed revisions. Manuscript is now suitable for publication in Crystals